# Effects of Vegetation Restoration on the Distribution of Nutrients, Glomalin-Related Soil Protein, and Enzyme Activity in Soil Aggregates on the Loess Plateau, China

**Leilei Qiao** [1,2], **Yuanze Li** [3], **Yahui Song** [2], **Jiaying Zhai** [2], **Yang Wu** [3], **Wenjing Chen** [3], **Guobin Liu** [1,2] and **Sha Xue** [2,3,4,*]

1   State Key Laboratory of Soil Erosion and Dryland Farming on Loess Plateau, Institute of Soil and Water Conservation, Northwest A&F University, Yangling 712100, China
2   Institute of Soil and Water Conservation, Chinese Academy of Sciences and Ministry Water Resources, Yangling 712100, China
3   College of Forestry, Northwest A&F University, Yangling 712100, China
4   Shaanxi Key Laboratory of Land Consolidation, Chang'an University, Xi'an 710000, China
*   Correspondence: xuesha100@163.com; Tel.: +1-367-921-5717

**Abstract:** Research Highlights: Soil enzymes have a significant impact on the production of glomalin-related soil protein (GRSP), directly and indirectly affecting the nutrient metabolism balance, but there is little available information on ecological stoichiometry in soil aggregates. Background and Objectives: Vegetation restoration changes community structure and species composition in ecosystems, thus changing the physicochemical properties of soil. Soil aggregate is the most basic physical structure of the soil. Therefore, in order to understand dynamic changes in soil aggregate nutrients as vegetation restoration progresses, we set out to investigate the nutrient distribution and utilization in aggregates, and how enzymes respond to the nutrient changes in achieving a nutritional balance along successive stages of vegetation restoration. Materials and Methods: We collected and analyzed soil from plots representing six different stages of a vegetation restoration chronosequence (0, 30, 60, 100, 130, and 160 years) after farmland abandonment on the Loess Plateau, China. We investigated soil nutrient stoichiometry, GRSP, and enzyme stoichiometry in the different successional stages. Results: The results revealed that soil organic carbon, total nitrogen, enzyme activity, and GRSP increased with vegetation recovery age, but not total phosphorus, and not all enzymes reached their maximum in the climax forest community. The easily extractable GRSP/total GRSP ratio was the largest at the shrub community stage, indicating that glomalin degradation was the lowest at this stage. Ecological stoichiometry revealed N-limitation decreased and P-limitation increased with increasing vegetation restoration age. Soil enzymes had a significant impact on the GRSP production, directly and indirectly affecting nutrient metabolism balance. Conclusions: Further study of arbuscular mycorrhizal fungi to identify changes in their category and composition is needed for a better understanding of how soil enzymes affect their release of GRSP, in order to maintain a nutrient balance along successive stages of vegetation restoration.

**Keywords:** ecological stoichiometry; nutrient limitation; soil enzymes; arbuscular mycorrhizal fungi

## 1. Introduction

Vegetation restoration improves the physicochemical properties and microbial composition of soil through changes in community structure and species types [1]. It is also conducive to the restoration of

degraded soils, such as those of the Loess Plateau in China, where soil degradation has historically been particularly severe due to agricultural expansion, deforestation, and resultant desertification [2,3]. Previous studies have shown an increase in nutrient content, and the microbial and enzyme activity due to inputs from animal and plant residues and root hyphae, as well as decreased soil bulk density and increased soil aggregate stability with increasing vegetation restoration age [4–6].

Soil aggregates have a major impact on properties of soil influencing its basic physical structure, including soil porosity, organic carbon (OC) stabilization, hydraulic conductivity, water-holding capacity, and soil erosion resistance [7–9]. The physical protection provided to OC by soil aggregates is conducive to carbon sequestration [10,11]. Additionally, a stable aggregate structure provides a suitable environment for microbes to perform the biogeochemical reactions that drive ecosystem functioning [12]. Glomalin, a glycosylated protein, which is produced in soil by arbuscular mycorrhizal fungi (AMF), promotes the stability of aggregates by a gluing action [13]. Glomalin in the soil is always quantified as glomalin-related soil protein (GRSP), which is generally divided into two fractions: total GRSP (GRSP-T) and easily extractable GRSP (GRSP-EE) [13–15]. GRSP promotes the accumulation of C and N by retaining soil C and N [16,17] and can be used as a binder to improve the stability of soil aggregates [13]. Therefore, GRSP is considered to be an important indicator in monitoring soil degradation [18,19]. In order to monitor changes in soil quality during vegetation restoration, it is essential to understand the cumulative dynamics of soil nutrients and glomalin and to explore the mechanisms underlying aggregate stability. In this context, constant efforts have been made to explore ways of improving the ecological environment and soil quality of the Loess Plateau, due to its degradation and inherently fragile characteristics, such as its highly erodible and poor soil.

Soil microorganisms promote the decomposition of organic matter by releasing enzymes to alleviate nutrient limitation. For example, the expression of C, N, and P-acquiring enzymes is a product of cellular metabolism specifically regulated by environmental nutrient availability [20]. Ecoenzymatic stoichiometry is a useful indicator of microbial combinations and their relative resource constraints, as extracellular enzyme activity reflects the response of microbial cells to meet metabolic resource demand [20,21]. Vegetation restoration has a significant impact on enzyme activity. Zhao et al. (2018) found that β-1, 4-glucosidase (BG), alkali phosphatase (AP), and β-N-acetyl glucosaminidase (NAG) activity was higher on afforested land on the Loess Plateau, and increased with increasing time post-afforestation when compared with farmland on which afforestation had been abandoned [22]. Most studies and meta-analyses have indicated a tendency for the ratio of enzymes involved in C, N, and P cycling to be 1:1:1, and C-, N-, and P-limitation are common among microbes [23]. Based on a meta-analysis of enzyme activities, Waring et al. (2014) found that P-limitation in tropical forests was reflected in lower BG:AP and NAG:AP ratios [24]. However, high-latitude forests tend to be N-limited [25]. This phenomenon is mainly caused by the high weathering rate of tropical forests leading to P loss, and episodic glaciations in high-latitude forests leading to limitation of N accumulation [26,27]. This implies that soil extracellular enzyme stoichiometry is mainly controlled by soil C, N, and P stoichiometry. Moreover, potential enzyme activities are normalized by microbial biomass C and soil total C to obtain two different specific activity indices, which are both beneficial in exploring changes in enzyme activity per unit of soil total C (microbial biomass C). Research by Peng et al. (2016) found that soil extracellular enzyme activity (EEA) was higher in meadow steppes when compared with typical and desert steppes, and declined from the topsoil to subsoil. However, when normalized by soil C and microbial biomass C, soil EEA significantly increased throughout the soil profile [28]. Soil aggregates enable the physical protection and isolation of nutrients and changes to sites of microbial activity which may affect nutrient distribution and utilization. Therefore, it is imperative to investigate the relationship between nutrient stoichiometry and enzyme stoichiometry in soil aggregates to reveal the nutrient limitation mechanisms operating during vegetation restoration.

Our study investigated the distribution characteristics of soil aggregate nutrients, GRSP, and enzyme activity following the restoration of the natural vegetation on the Loess Plateau in China. Although dynamic changes in nutrients, microorganisms, and enzyme activity with increasing

vegetation restoration age have been well studied in this region [29–32], there is little available information on ecological stoichiometry in soil aggregates. Our main objectives were therefore to explore the effect of vegetation restoration on GRSP, nutrients, and enzyme activity in aggregates, and to identify nutrient limitation in soil aggregates at different stages using stoichiometry. Specifically, we hypothesized that: (1) soil nutrients, GRSP, and enzyme activity increase as vegetation restoration age increases, (2) levels of P-limitation will increase and N-limitation decrease along a vegetation restoration chronosequence, and (3) soil enzymes promote the secretion of GRSP by AMF, affecting the balance of soil carbon and nitrogen metabolism.

## 2. Materials and Methods

### 2.1. Study Area and Site Selection

The sampling site was located in the Lianjiabian Forest Farm in the Ziwuling Forest Region of the Loess Plateau, China (35°03′–36°37′ N, 108°10′–109°18′ E). This region has a temperate continental monsoon climate, with a mean annual temperature of 10 °C and mean annual precipitation of 587 mm. The natural biomes are deciduous broadleaf forests: *Populus davidiana* Dode. and *Quercus liaotungensis* Koidz. dominate the region's pioneer and climax forests, respectively [32]. The region is the foothill-gully zone of the Loess Plateau, with an average elevation of 1500 m [33–35] (Figure 1).

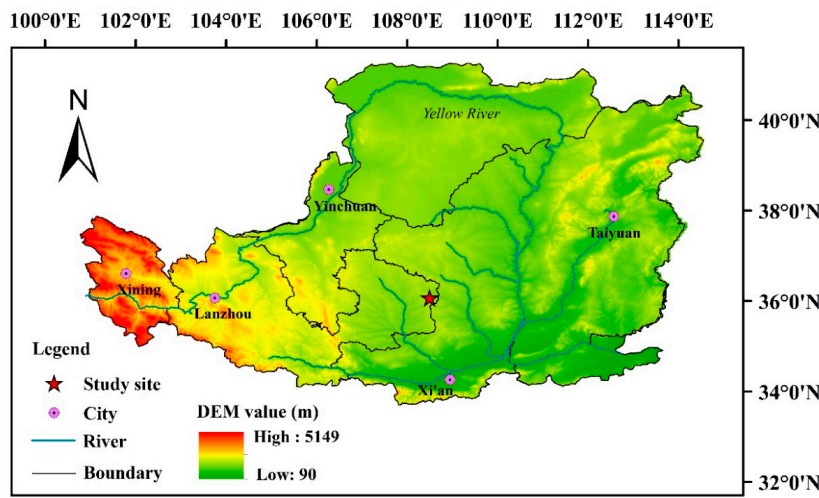

**Figure 1.** Location of the study site (Ziwuling Forest Region) on the Loess Plateau, China. DEM, digital elevation model.

The secondary succession naturally regenerated on abandoned cropland after many local inhabitants emigrated from the Ziwuling Forest Region during a national conflict in 1842–1866. According to previous reports, natural restoration of vegetation from farmland to *P. davidiana* and *Q. liaotungensis* forests takes about 100 and 160 years, respectively [35,36]. In the 1940s to 1960s, the Ziwuling region was affected by the settlement of people who reclaimed land due to famine, war, and disasters. The arable land has been abandoned more than once in different areas in the process of human emigration and immigration, a secondary succession of vegetation in abandoned tillage has led to the existence of vegetation at different stages of restoration in the region [37]. For our study, we selected six communities that had undergone vegetation restoration for about 0, 30, 60, 100, 130, and 160 years since farmland abandonment, corresponding respectively to the farmland, herbaceous community, shrub community, pioneer forest, mingled forest, and climax forest community stages. Two methods were used to verify the community ages (Table 1). For the shrub and herbaceous communities (<60 years of succession), we verified the community age both by visiting local elders and referring to land contracts between local elders and the government. We determined the ages of the forest communities (>60 years post-recovery) by boring tree rings and checking related written sources [37].

**Table 1.** Geographical and vegetation characteristics at different stages of restoration in the Ziwuling forest region of the Loess Plateau, China.

| Site | Succession Stage (in Years) | Biome | Altitude (m) | Slope (°) | Main Plant Species |
|------|------|------|------|------|------|
| S1 | 0-y | Farmland | 1280 | 0 | *Corn* |
| S2 | 30-y | Herbaceous | 1416 | 15 | *Bothriochloa ischaemum, Lespedeza dahurica* |
| S3 | 60-y | Shrub | 1346 | 21 | *Sophora davidii, Carex lanceolata, Hippophae rhamnoides* |
| S4 | 100-y | The pioneer forest | 1445 | 14 | *Populus davidiana, Spiraea schneideriana, Carex lanceolata* |
| S5 | 130-y | The mingled forest | 1440 | 18 | *Populus davidiana, Quercus liaotungensis, Betula platyphylla* |
| S6 | 160-y | The climax forest | 1427 | 22 | *Quercus liaotungensis, Rosa hugonis, Carex lanceolata* |

## 2.2. Soil Sampling

Soil samples (undisturbed soil) were collected from Lianjiabian Forest Farm on September 10, 2017. Four sampling plots with similar slopes (<20°), aspects, and altitudes were randomly chosen from each of the six vegetation communities, with a distance between adjacent plots of at least 80–100 m. The sizes of sample plots in the forest, shrub, and herbaceous communities were $20 \times 20$ m, $5 \times 5$ m, and $2 \times 2$ m, respectively. The surface layer (0–5 cm) of bulk soils were collected in quadruplicate from each sample plot using a soil core sampler and mixed to form one sample. Before the soil collection, litter and debris on the ground surface were removed. Disturbance of the soil was avoided during its subsequent transportation to avoid destroying its physical structure. The soil samples were then dried to approximately 10% gravimetric water content in a 4 °C environment to ensure that environmental changes in the soil sample were minimized without affecting the screening [12]. The soil samples were then gently sorted by hand to below 8 mm in size, and sieved for 4 min on a mechanical shaker to partition the aggregate sizes. Soil samples were sieved into three grades using stacking sieves (2 and 0.25 mm): >2 mm (large macroaggregates), 2–0.25 mm (small macroaggregates), and <0.25 mm(microaggregates) [38]. All visible gravel and roots were removed, a part of the soil sample was air-dried prior to measuring its physicochemical properties, and the other part was placed in a refrigerator at −20 °C to measure enzyme activity. The soil samples at −20 °C were thawed to 4 °C three days prior to the start of the soil enzyme activity assay.

## 2.3. Soil Physicochemical Properties and Enzyme Analysis

Soil OC (SOC) was measured using the dichromate oxidation method [39], soil total nitrogen (TN) was measured using the Kjeldahl method [40], and soil total phosphorus (TP) was determined colorimetrically after digestion with $H_2SO_4$ and $HClO_4$ [41]. GRSP was extracted from soil samples using the procedures described by Wright and Upadhyaya (1996) for GRSP-EE and GRSP-T [42].

The potential activities of two C-acquiring enzymes (BG, BX), three N-acquiring enzymes (NAG: leucine aminopeptidase, LAP, ALT: alanine-$\alpha$-aminopeptidase), and one P-acquiring enzyme (AP) were measured by following modified versions of standard fluorometric techniques [43,44]. Briefly, 1.0 g of a soil sample was placed in 125 mL (pH = 8.5) sodium acetate buffer and shaken for 1 h. We used a 96-well microplate for detection, with eight replicate microwells per analysis. The analysis includes a sample reaction microwell (150 μL sample suspension + 50 μL fluorometric substrate), blank microwell (150 μL sample suspension + 50 μL buffer), negative control microwell (150 μL buffer + 50 μL fluorometric substrate), quench standard microwell (150 μL sample suspension + 50 μL standard), and reference standard microwell (150 μL buffer + 50 μL standard). After the microplate was loaded, the mixture was shaken and mixed uniformly, and cultured in a 25 °C incubator for 2 or 4 h. The amount of fluorescence was determined using a microplate reader (365 nm for excitation and 450 nm for emission). After correcting the negative control and quench standard, enzyme activity was measured in units of $nmol \cdot g^{-1} \cdot h^{-1}$.

## 2.4. Data Analysis

In our study, one-way analysis of variance (ANOVA) was used to analyze the effect of vegetation restoration on the distribution of GRSP, SOC, TN, TP, and enzyme activity in soil aggregates. Duncan's tests at $p < 0.05$ were carried out for multiple comparisons. SPSS version 20.0 (IBM SPSS, Chicago, IL, USA) was used for all statistical analyses, and graphs were drawn using Origin 9.0 (OriginLab Corporation, Northampton, MA, USA).

A structural equation model (SEM) was constructed based on the effects of vegetation restoration and soil aggregate size on nutrients, GRSP, and ecoenzymatic stoichiometry. We first reduced the number of variables for the GRSP content and enzyme stoichiometry using principal component analysis (PCA) [45,46]. GRSP-EE and GRSP-T content (GRSP) and C-, N-, and P-acquiring enzyme activity (ecoenzymatic stoichiometry) were used as raw data in the PCA. The first principal component (PC1) was used in the subsequent SEM analysis to represent GRSP concentration (PC1 explained 85.781% of the variance) and ecoenzymatic stoichiometry (PC1 explained 63.162% of the variance). The SEM analysis was performed using the AMOS software package (version 21.0, IBM SPSS, IBM SPSS Corporation, Chicago, IL, USA manufacturer, city and country).

## 3. Results

### 3.1. Aggregate Stability and Content of Nutrients and GRSP

With the progressive restoration of vegetation, the mean weight diameter (MWD) of soil aggregates increased, indicating an increase in the degree of aggregate polymerization (Figure 2).

The SOC and TN of different aggregate sizes increased as vegetation restoration progressed, reaching their maximum at 160 y (Figure 3a,b). In plots representing farmland or farmland 30-y post-abandonment, SOC and TN values were significantly lower ($p < 0.05$) than in forest (100-y age), and were also significantly lower in 60-y compared with 160-y communities ($p < 0.05$) (Figure 3a,b). SOC and TN accumulated rapidly from 30-y to 100-y post-restoration, after which the growth rate stabilized and was slower (Figure 3a,b). The highest TP content was found in farmland, and there was no significant difference in TP with increasing vegetation restoration age (Figure 3c). There was no significant difference in SOC, TN, and TP for different aggregate sizes, except for SOC in the farmland stage (Figure 3a–c).

GRSP-EE and GRSP-T content increased rapidly 30–60 y and 60–100 y, respectively, after vegetation restoration, and then increased slowly from 100-y after farmland abandonment (Figure 4a,b). The ratio of GRSP-EE/GRSP-T achieved the largest value at 60-y and was stable in the other stages between 0.1 and 0.16 (Figure 4c). No significant difference was found between different sizes of soil aggregate.

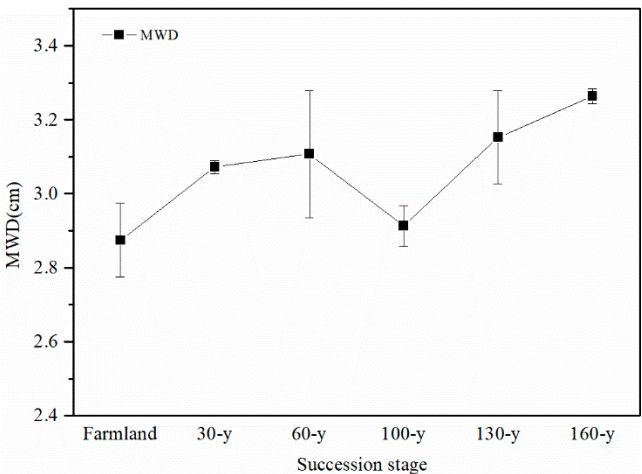

**Figure 2.** Changes in MWD in Surface layer with vegetation succession.

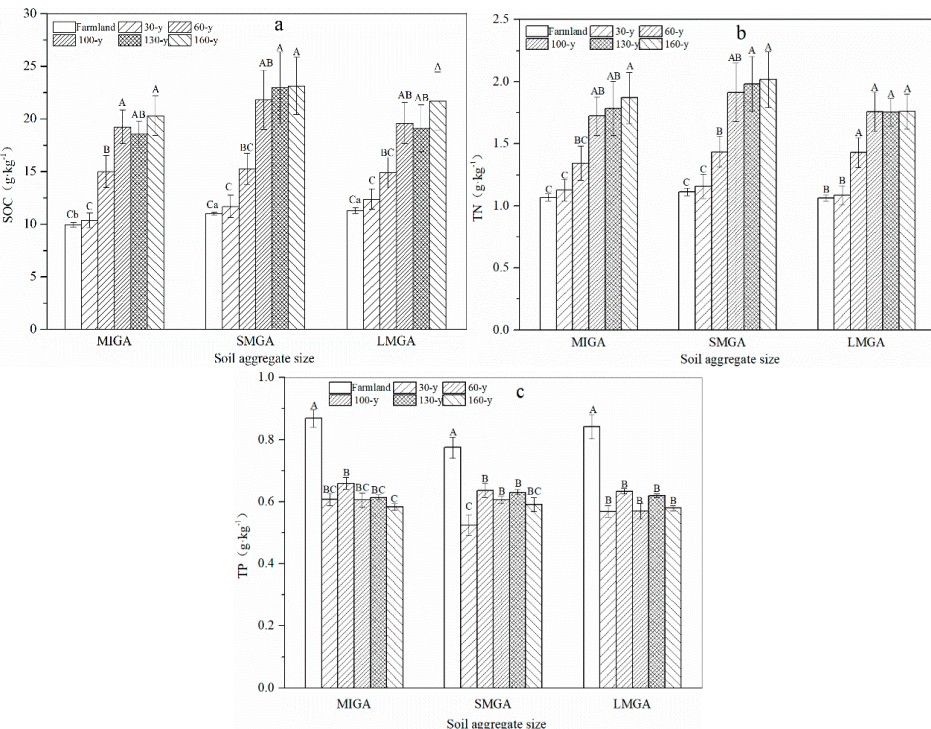

**Figure 3.** Changes in soil organic carbon (SOC) (**a**), total nitrogen (TN) (**b**), and total phosphorus (TP) (**c**) in different sizes of soil aggregates with the vegetation succession. MIGA: microaggregate, SMGA: small macroaggregate, LMGA: large macroaggregate. Error bars denote Standard error. Letters above the bars are for comparison in the size of the same aggregate at the different restoration stages, bars with the same letter are not significantly different at *p* = 0.05.

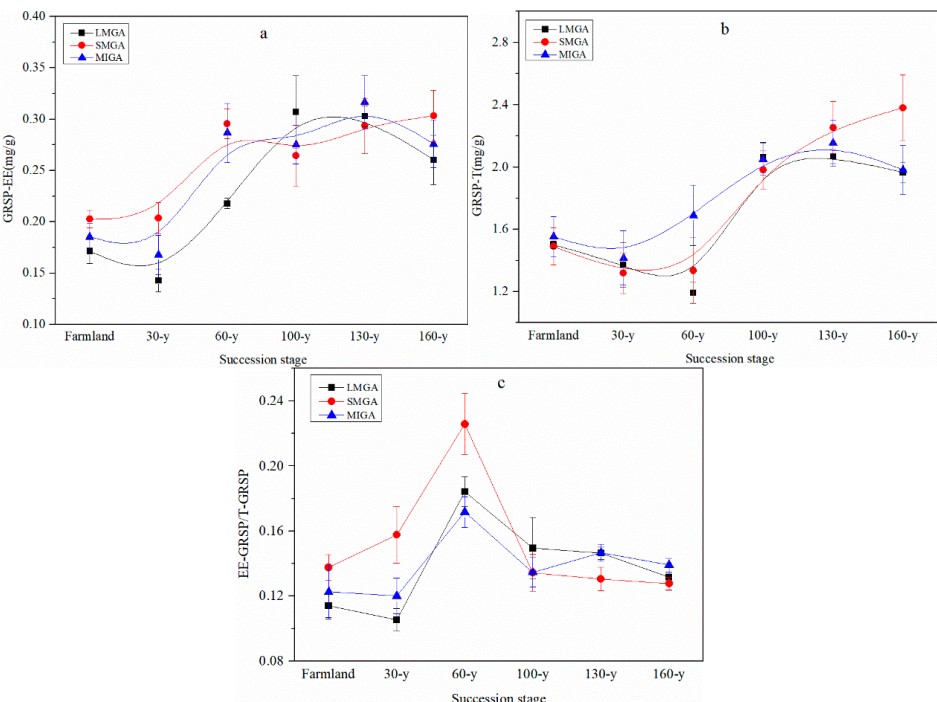

**Figure 4.** Changes in easily extractable GRSP (GRSP-EE) (**a**), total GRSP (GRSP-T) (**b**), and GRSP-EE/GRSP-T (**c**) in different sizes of soil aggregates with the vegetation succession. Error bars denote Standard error.

## 3.2. Enzyme Activity and Nutrient Turnover

The activity of C-acquiring enzymes (BG and BX) showed an increasing trend after vegetation restoration, mainly divided between the two stages of 0–60 y and 100–160 y, but was lower than either of these stages at 130-y (Figure 5a,b). LAP and ALT activity was divided into two stages, reaching maximum values at 60-y and 160-y as vegetation restoration progressed, respectively (Figure 5c,d). NAG activity increased with increasing community age after vegetation restoration and was significantly lower at 0-60y compared with 160-y (Figure 5e). AP activity was less affected by vegetation restoration age. It increased slowly with increasing recovery age and tended to be stable after 100 y (Figure 5f). There was no significant difference in C-, N-, and P-acquiring enzyme (BG, BX, LAP, ALT, NAG, and AP) activities among the different soil aggregate sizes.

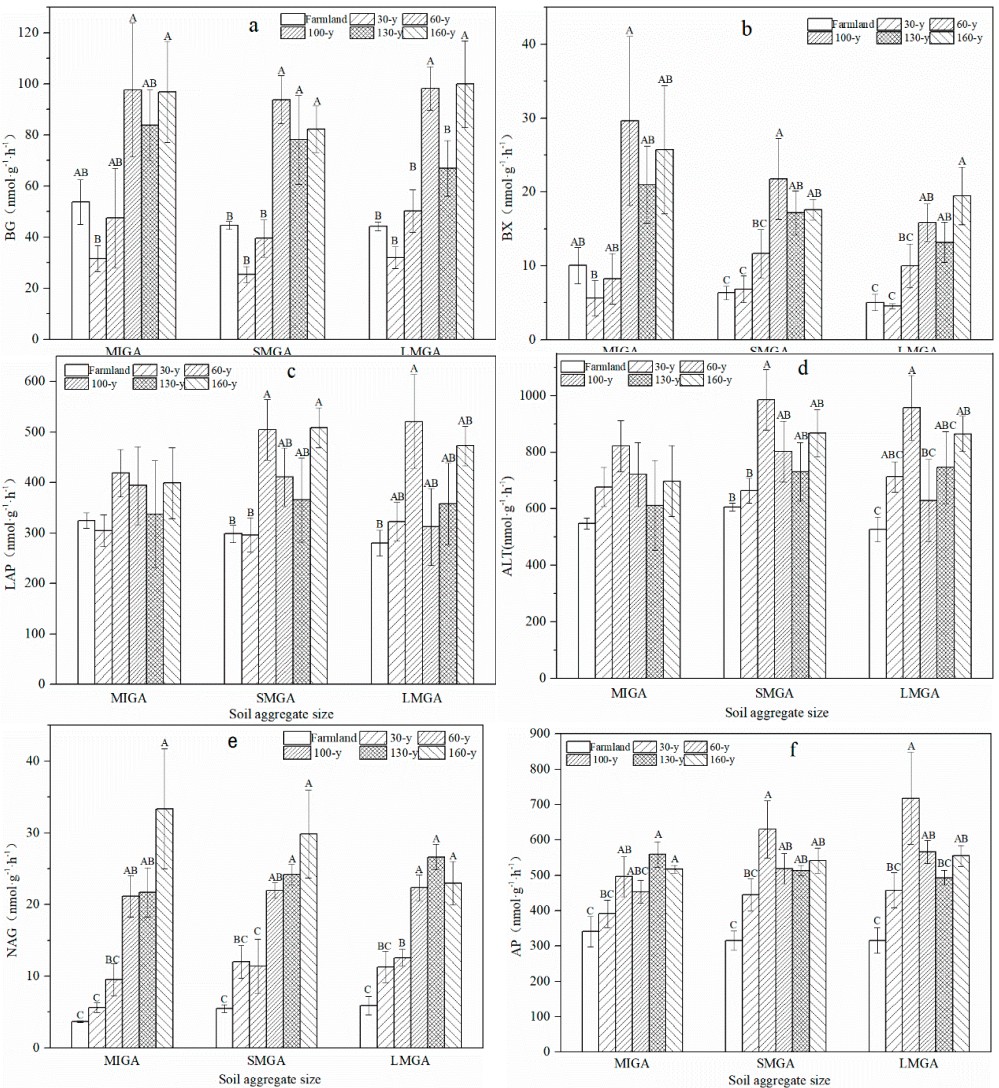

**Figure 5.** Changes in BG (**a**), BX (**b**), NAG (**c**), LAP (**d**), ALT (**e**), and AP (**f**) in different sizes of soil aggregates with the vegetation succession. Error bars denote Standard error. Letters above the bars are for comparison in the size of the same aggregates at the different restoration stages, bars with the same letter are not significantly different at *p* = 0.05. BG: β-1, 4-glucosidase, BX: β-1,4-xylosidase, NAG: β-N-acetyl glucosaminidase, LAP: leucine aminopeptidase, ALT: alanine-α-aminopeptidase, AP: alkali phosphatase.

The enzyme activity per nutrient unit was quantified using the ratio of enzyme activity to corresponding nutrient content to reveal the nutrient turnover rate at different succession stages. C-acquiring enzymes achieved the largest ratio at the 100-y stage and then stabilized (Figure 6a). The ratio for N-acquiring enzymes gradually decreased with progressive succession stages and tended to be stable after 130 y (Figure 6b). The ratio of P-acquiring enzymes increased with increasing restoration age until 100 y and then stabilized (Figure 6c).

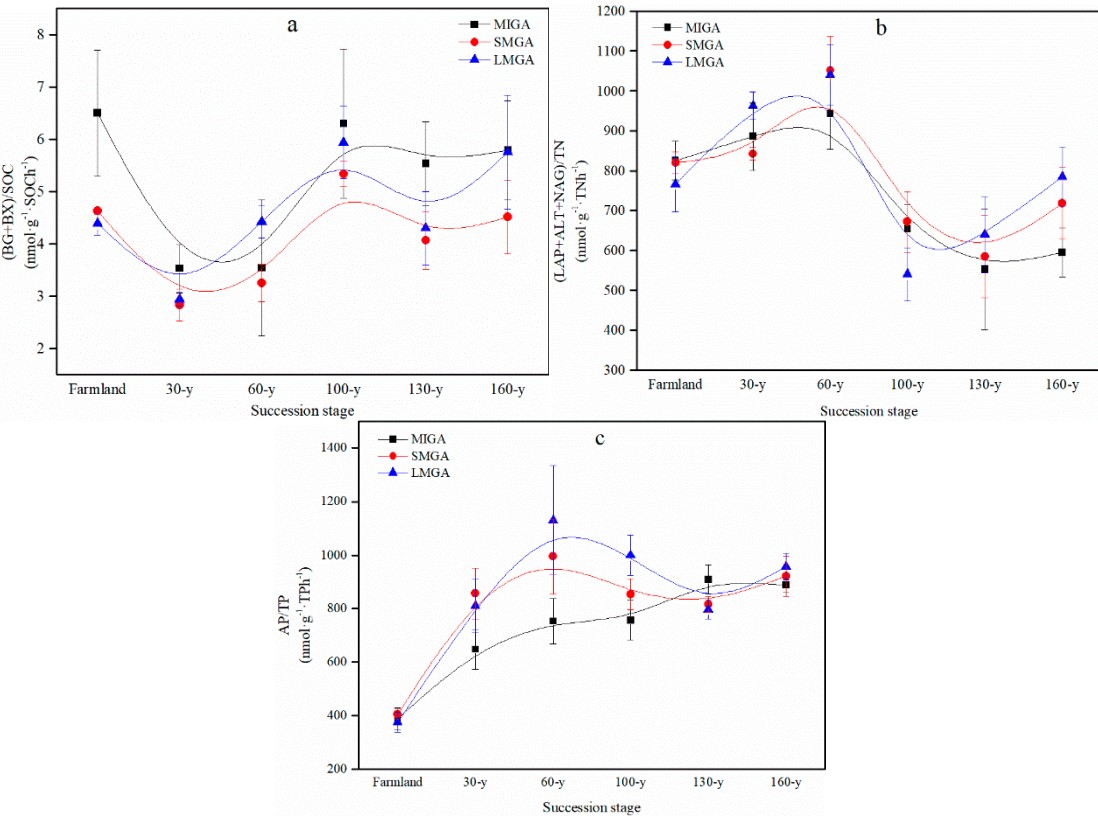

**Figure 6.** Changes in (BG+BX)/SOC (**a**), (NAG+ LAP+ ALT)/TN (**b**), and AP/TP (**c**) in different sizes of soil aggregates with the vegetation succession. Error bars denote Standard error.

### 3.3. Ecological Stoichiometry

The C: N, C:P, and N:P values in different sizes of soil aggregates increased with increasing vegetation restoration age. Soil C:P and N:P showed no significant difference among aggregates of different sizes (Table 2). Soil C:P and N:P were largest in SMGA and Soil C:P increased following sequence: MIGA < LMGA < SMGA. (Table 2). C: N values in MIGA were significantly lower than in LMGA in farmland and 30-y succession stages (Table 2).

Enzyme C: N, C: P, and N: P ratios showed no significant difference among different aggregate sizes. Enzyme C: N and C: P ratios showed an increasing trend with increasing vegetation restoration age, the enzyme N:P ratio showed a decreasing trend, and there was no significant difference among MIGA as vegetation restoration progressed (Table 3).

**Table 2.** Effects of different particle size aggregates on topsoil stoichiometry in different succession stages.

| Succession Stage | Soil C: N Ratio | | | Soil C: P Ratio | | | Soil N: P Ratio | | |
|---|---|---|---|---|---|---|---|---|---|
| | MIGA | SMGA | LMGA | MIGA | SMGA | LMGA | MIGA | SMGA | LMGA |
| Farmland | 9.299 Bb | 9.894 Bab | 10.604 a | 11.471 C | 14.285 B | 13.518 C | 1.230 B | 1.437 C | 1.264 D |
| 30-y | 9.221 Bc | 10.086 Bb | 11.379 a | 17.016 BC | 22.498 B | 21.662 BC | 1.855 B | 2.205 BC | 1.908 CD |
| 60-y | 11.206 A | 10.597 B | 10.428 | 22.683 B | 24.002 B | 23.425 BC | 2.037 B | 2.258 BC | 2.257 BC |
| 100-y | 11.185 A | 11.374 A | 11.124 | 31.971 A | 35.748 A | 34.762 A | 2.844 A | 3.155 AB | 3.081 A |
| 130-y | 10.661 AB | 11.449 A | 10.854 | 30.181 A | 36.438 A | 30.810 AB | 2.905 A | 3.144 AB | 2.829 AB |
| 160-y | 10.955 A | 11.439 A | 12.214 | 34.834 A | 39.810 A | 37.676 A | 3.206 A | 3.415 A | 3.036 A |

**Table 3.** Effects of different particle size aggregates on the enzymatic stoichiometry of topsoil in different succession stages.

| Succession Stage | Enzymatic C: N Ratio | | | Enzymatic C: P Ratio | | | Enzymatic N: P Ratio | | |
|---|---|---|---|---|---|---|---|---|---|
| | MIGA | SMGA | LMGA | MIGA | SMGA | LMGA | MIGA | SMGA | LMGA |
| Farmland | 0.077 AB | 0.055 BC | 0.062 BCD | 0.191 AB | 0.165 A | 0.161 A | 2.811 | 3.089 A | 2.740 A |
| 30-y | 0.037 B | 0.034 C | 0.035 D | 0.098 B | 0.074 B | 0.083 B | 2.624 | 2.220 B | 2.367 AB |
| 60-y | 0.044 B | 0.034 C | 0.045 CD | 0.110 B | 0.082 B | 0.099 B | 2.539 | 2.413 AB | 2.169 AB |
| 100-y | 0.105 A | 0.094 A | 0.126 A | 0.272 A | 0.222 A | 0.200 A | 2.489 | 2.439 AB | 1.673 B |
| 130-y | 0.119 A | 0.088 AB | 0.072 BC | 0.185 AB | 0.188 A | 0.163 A | 1.727 | 2.179 B | 2.303 AB |
| 160-y | 0.107 A | 0.071 AB | 0.086 B | 0.237 AB | 0.190 A | 0.215 A | 2.170 | 2.632 AB | 2.453 AB |

Note: Different capitals in the same column indicate a significant difference in different successional stages at $p < 0.05$, The same line with different lowercase letters indicates significant difference among aggregates with different diameters at $p < 0.05$.

*3.4. Comprehensive Response of Soil Aggregate Nutrients and Enzymes to Vegetation Restoration*

Correlation analysis demonstrated GRSP-EE and GRSP-T had a significantly positive correlation with SOC, TN, C: N, C: P, N: P, enzymatic C: N, and enzymatic C: P, except for enzymatic N: P. GRSP-EE was significantly positively correlated with all enzymes. In addition to AP and ALT, GRSP-T was significantly positively correlated with other enzymes. GRSP-EE and GRSP-T had a negative correlation with TP (Table 4).

The SEM fitted the data well and revealed the response of enzyme interactions, soil nutrient content, and GRSP content to vegetation restoration. Vegetation restoration increased SOC, TN, GRSP content, and enzyme activity. GRSP had a significant effect on increases in SOC and TN content. Soil enzymes had a significant impact on GRSP production, directly and indirectly affecting SOC and N accumulation. Soil aggregate size had a significant effect on SOC but was not significantly correlated with other indicators (Figure 7).

**Table 4.** Correlation coefficients of contents and stoichiometry between soil properties and ecoenzymes.

| | C | N | P | C:N | C:P | N:P | BG | BX | ALT | LAP | NAG | AP | Enzymatic C:N | Enzymatic C:P | Enzymatic N:P | GRSP-EE | GRSP-T |
|---|---|---|---|---|---|---|---|---|---|---|---|---|---|---|---|---|---|
| C | 1.00 | | | | | | | | | | | | | | | | |
| N | 0.947 ** | 1.00 | | | | | | | | | | | | | | | |
| P | −0.387 ** | −0.363 ** | 1.00 | | | | | | | | | | | | | | |
| C:N | 0.620 ** | 0.350 ** | −0.302 ** | 1.00 | | | | | | | | | | | | | |
| C:P | 0.968 ** | 0.915 ** | −0.582 ** | 0.613 ** | 1.00 | | | | | | | | | | | | |
| N:P | 0.917 ** | 0.953 ** | −0.603 ** | 0.387 ** | 0.962 ** | 1.00 | | | | | | | | | | | |
| BG | 0.739 ** | 0.783 ** | −0.17 | 0.258 * | 0.693 ** | 0.719 ** | 1.00 | | | | | | | | | | |
| BX | 0.679 ** | 0.772 ** | −0.21 | 0.13 | 0.650 ** | 0.720 ** | 0.820 ** | 1.00 | | | | | | | | | |
| ALT | 0.311 ** | 0.340 ** | −0.23 | 0.11 | 0.307 ** | 0.334 ** | 0.299 * | 0.346 ** | 1.00 | | | | | | | | |
| LAP | 0.417 ** | 0.473 ** | −0.18 | 0.11 | 0.415 ** | 0.460 ** | 0.397 ** | 0.520 ** | 0.713 ** | 1.00 | | | | | | | |
| NAG | 0.751 ** | 0.747 ** | −0.468 ** | 0.418 ** | 0.770 ** | 0.773 ** | 0.628 ** | 0.624 ** | 0.263 * | 0.331 ** | 1.00 | | | | | | |
| AP | 0.425 ** | 0.428 ** | −0.385 ** | 0.261 * | 0.448 ** | 0.459 ** | 0.289 * | 0.327 ** | 0.523 ** | 0.613 ** | 0.320 ** | 1.00 | | | | | |
| enzymatic C:N | 0.588 ** | 0.624 ** | −0.06 | 0.20 | 0.534 ** | 0.552 ** | 0.815 ** | 0.653 ** | -0.21 | -0.04 | 0.480 ** | 0.11 | 1.00 | | | | |
| enzymatic C:P | 0.569 ** | 0.636 ** | 0.04 | 0.09 | 0.500 ** | 0.537 ** | 0.898 ** | 0.783 ** | 0.13 | 0.23 | 0.503 ** | −0.09 | 0.780 ** | 1.00 | | | |
| enzymatic N:P | −0.100 | −0.04 | 0.20 | −0.23 | −0.13 | −0.10 | 0.04 | 0.09 | 0.441 ** | 0.300 * | −0.06 | −0.444 ** | −0.305 ** | 0.286 * | 1.00 | | |
| GRSP-EE | 0.780 ** | 0.819 ** | −0.273 * | 0.322 ** | 0.735 ** | 0.761 ** | 0.606 ** | 0.595 ** | 0.269 * | 0.381 ** | 0.637 ** | 0.407 ** | 0.533 ** | 0.457 ** | −0.12 | 1.00 | |
| GRSP-T | 0.794 ** | 0.819 ** | −0.23 | 0.336 ** | 0.759 ** | 0.771 ** | 0.720 ** | 0.602 ** | 0.11 | 0.277 * | 0.698 ** | 0.13 | 0.642 ** | 0.669 ** | 0.06 | 0.716 ** | 1.00 |

Note: Significant correlations are indicated by asterisks (*) ($p < 0.05$); Extremely significant correlations are indicated by asterisks (**) ($p < 0.01$).

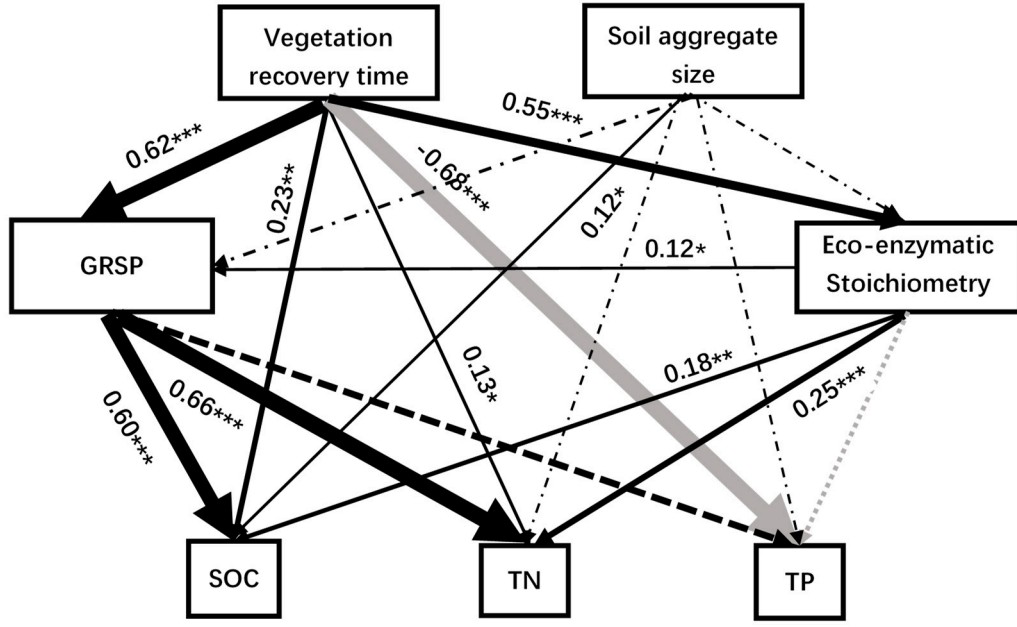

$\chi^2$=0.447；DF=2；P=0.800；GFI=0.998；AIC=52.447；RMSEA=0.000

**Figure 7.** Structural equation model of the effects of vegetation restoration on soil aggregate nutrients, GRSP, and Eco-enzymatic Stoichiometry. Solid arrows indicate positive relationships, gray arrows indicate negative relationships and dashed arrows indicate no significant correlation. *, $p < 0.05$, **, $p < 0.01$, ***, $p < 0.001$.

## 4. Discussion

### 4.1. Effects of Vegetation Restoration on Soil Aggregate Nutrients and Glomalin

The results of our study were in agreement with previous studies in showing that soil aggregate MWD values increased with increasing successional age [47,48], indicating that vegetation restoration has a significant effect on soil improvement and development.

SOC is mainly derived from vegetation litter, root exudates, and the remains of animals and plants [48,49]. With vegetation restoration and succession, the vegetation types at the surface changed from herbs to a forest, and the increase in litter entering the soil promotes SOC and TN accumulation. After forestation, vegetation type changed little and SOC and TN content increased slowly. The maximum value for SOC and TN content was found in the climax forest community stage, as reflected in our study results. TP content was mainly influenced by the parent material, vegetation type, and soil biogeochemical processes, and revealed a large spatial heterogeneity [50,51]. As the soil parent material and climate conditions of all the vegetation types were similar, the age of vegetation restoration had little effect on TP content, the farmland was instead mainly affected by artificial fertilization.

Glomalin content is positively correlated with net primary production (NPP), which determines the upper limit of SOC available for glomalin production and turnover rate of AMF [52,53]. Therefore, glomalin accumulates more quickly where NPP is greater [53,54]. In our study, it was found that GRSP-EE and GRSP-T contents increased with increasing vegetation restoration age, growing rapidly from the herb to shrub community stage and from the shrub to pioneer forest community stage, respectively. This may be due to changes in species dominant in the vegetation restoration process affecting the cumulative GRSP rate. GRSP-EE is considered to be a recently produced fungal protein, while GRSP-T is the sum of both recent and past fungal protein production [15,42]. Therefore, GRSP-EE is more sensitive, and the rapid accumulation phase of GRSP-T occurs later than GRSP-EE due to hysteresis. The GRSP-EE/GRSP-T ratio can reflect differences in degradation rates and the potential

increase in soil GRSP [55]. In our study, GRSP-EE accounted for the highest proportion of GRSP-T in the shrub community stage, indicating that glomalin degradation was the lowest at the shrub community stage, which was conducive to SOC accumulation and storage.

### 4.2. Effects of Vegetation Restoration on Soil Aggregate Enzyme Activity and Nutrient Turnover

Previous research has shown that soil enzyme activity is closely related to SOC because the transformations of important organic elements are facilitated by microorganisms [56,57]. With increasing age of vegetation restoration, aboveground and underground litter increased, and soil organic matter was abundant, which further stimulated the synthesis of soil enzymes, and improved enzyme activity [5]. The results of our study agreed with this finding. We also found that ALT and LAP showed the highest activity at the shrub community stage, possibly related to the type of vegetation. The shrub community stage vegetation is dominated by leguminous plants, such as *Sophora davidii* (Franch.) Skeels. and *Hippophae rhamnoides* Linn., so ALT and LAP activity is higher [30]. LAP and ALT hydrolyze the most abundant protein amino acid from polypeptide ends. NAG, which hydrolyzes N-acetylglucosamine from chitobiose and other chito-oligosaccharides, is the most commonly measured indicator enzyme for chitin. According to previous research, which showed that NAG activity is mainly expressed by a diverse group of fungi [58,59], the total number of fungi generally increases along the succession stages from farmland to forest [60]. AP hydrolyzes phosphate esters, including phosphomonoesters, phosphodiesters, and in some cases phosphosaccharides that release phosphate [61,62]. With increasing restoration age, the accumulation of organic matter stimulated AP secretion to a certain extent. After forestation, the cumulative rate decreased and AP activity tended to stabilize.

Soil EEA was normalized by SOC, TN, and TP in order to explore the turnover efficiency of soil nutrients. In our study, we found that C-acquiring enzyme activity per unit SOC reached its maximum at the pioneer forest community stage, indicating that C turnover efficiency was the highest at this stage. N-acquiring enzyme activity per unit TN decreased with increasing vegetation restoration age, indicating that the mineralization efficiency of N is weakened, therefore promoting N accumulation. Vegetation restoration age had little effect on TP turnover efficiency: it may be that soil TP was mainly related to the parent material and climate, and was less influenced by vegetation.

### 4.3. Response of Ecological Stoichiometry to Vegetation Restoration

The relationship between soil C: N and C: P with vegetation restoration age is directly related to the cumulative rate of soil C and N. In our study, C: N and C: P values increased with increasing vegetation restoration age, indicating that C accumulated faster than N and P under vegetation restoration conditions. We also found that N:P increased with increasing vegetation restoration age. As soil develops, the amount of P released by weathering decreases, because primary minerals are depleted [63]. Soil erosion also accelerates the loss of soil nutrients (especially phosphorus) on the Loess Plateau [64]. We found that soil C:P and N:P were the largest in SMGA, and soil C:P increased following sequence: MIGA < LMGA < SMGA. This implied that accumulation rates of C, N, and P differed among different aggregate sizes and C,N accumulate faster in SMGA.

The ratios of log-transformed C-, N-, and P-acquiring enzymes were 1:1.6:1.4 in the topsoil, which contrasts with the 1:1:1 ratio determined for the global ecosystem [23]. In our study, enzymatic C: N gradually increased with increasing vegetation restoration age, indicating that N limitation gradually decreased. Vegetation type shifted and more litter entered the soil as restoration age progressed, which greatly improved soil N content. This accelerated N accumulation and storage and reduced N limitation. The enzymatic N:P ratio gradually decreased, indicating that P limitation was strengthening and that microorganisms secreted more P-acquiring enzymes to meet growth requirements. TP is mainly affected by the parent material, and surface biomass increased with increasing vegetation recovery age, thus increasing the demand for P, resulting in a gradual increase in P limitation.

*4.4. The Effect of GRSP on Nutrient Balance*

Vegetation restoration promotes nutrient and GRSP accumulation, enhances enzyme activity, improves system self-regulation, and enhances resistance stability. As is generally understood, GRSP is a type of recalcitrant soil protein of the organic C and N reserved in soils [53,65]. It may have an impact on nutrient stoichiometry due to its contribution to C and N accumulation. In our study, we found through the SEM that GRSP had a significant effect on increases in SOC and TN content, and GRSP had a significantly positive correlation with SOC, TN, and its stoichiometry, which was verification of this point. Additionally, we discovered that soil enzymes affect GRSP production, have a direct and indirect effect on C and N accumulation and nutrient stoichiometry in maintaining a nutritional balance, according to the SEM. Soil aggregate size had no significant effect on other nutrient distribution and enzyme activity, except for SOC.

## 5. Conclusions

In this study, we investigated soil nutrient stoichiometry, GRSP and enzyme stoichiometry, and revealed nutrient limitation at different successional stages of vegetation restoration in the Ziwuling region of the Loess Plateau. We found that SOC and TN increased with vegetation recovery age, reaching a maximum in the climax forest community. GRSP-EE and GRSP-T increased with increasing age of vegetation restoration, and GRSP-EE accounted for the highest proportion of GRSP-T at the shrub community stage, which was conducive to SOC accumulation and storage in this stage. C-, N-, and P-acquiring enzyme activity increased with increasing vegetation restoration age. However, not all enzymes reached their maximum value in the climax forest community, so enzyme activity needs to be considered comprehensively when evaluating soil quality. Our research suggested that N-limitation was weakened and P-limitation gradually increased as vegetation restoration progressed. Soil enzymes have a significant impact on the production of GRSP, directly and indirectly affecting the nutrient metabolism balance. Our study contributes to a better understanding of the changes of soil nutrients and enzyme activity and ecological stoichiometry at an aggregate level along vegetation restoration on the Loess Plateau, China.

**Author Contributions:** The research design was completed by L.Q., S.X. and G.L. The manuscript was written by L.Q. The collection and analysis of soil samples were performed by L.Q., Y.L., Y.S., J.Z., Y.W., and W.C.

**Funding:** The research was funded by The National Key Research and Development Program of China(2016YFC0501707), National Natural Science Foundation of China (41771557) and the Fund Project of Shaanxi Key Laboratory of Land Consolidation (2019-JC15).

**Acknowledgments:** We thank Chunyu Bai for providing technical assistance in graphics production.

**Conflicts of Interest:** All the authors declare no conflicts of interest.

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
