# Peer review of "Effects of Vegetation Restoration on the Distribution of Nutrients, Glomalin-Related Soil Protein, and Enzyme Activity in Soil Aggregates on the Loess Plateau, China"

_forests, doi:10.3390/f10090796_

Round 1

Reviewer 1 Report

The article by Qiao et al. is well written and the data is properly collected and analyzed. Although soil functionality is thought to be related to the level of AMF glomalins, there is insufficient scientific evidence on the relationship between glomalins and soil physicochemical properties. Current data explains this well, and offers great progress in the related fields.

Minor comments

line 149, Does storage of soil samples at -20 degrees affect soil enzyme activity?

line 188, The meaning of MIGA, SMGA, and LMGA in Figure 3 should be explained here.

line 199, Figure 3c -> Figure 4c.

Author Response

Thank you very much for reviewing and commenting on the article.

(1) line 149, Does storage of soil samples at -20 degrees affect soil enzyme activity?

The soil samples at -20°C were thawed to 4°C three days prior to the start of any enzyme assay. The article lacks explanations for it, and it has been supplemented and improved in the article. The main reference papers are as follows:

[1] Wang R , Dorodnikov M , Yang S , et al. Responses of enzymatic activities within soil aggregates to 9-year nitrogen and water addition in a semi-arid grassland[J]. Soil Biology and Biochemistry, 2015.

[2] Wang, Y.L, Krogstad, T, Clarke, J.L, et al. Rhizosphere Organic Anions Play a Minor Role in Improving Crop Species' Ability to Take Up Residual Phosphorus (P) in Agricultural Soils Low in P Availability[J]. Frontiers in Plant Science. 2016, 7, 14.

(2) line 188, The meaning of MIGA, SMGA, and LMGA in Figure 3 should be explained here.

Thanks for your advice, it has been supplemented in article.

(3) line 199, Figure 3c -> Figure 4c.

Thanks for your advice, it has been modified in article.

Reviewer 2 Report

table and text include term 'migled forest' which is incomprehensible. Is this a regenerating forest, an artificial planting, a combination?

abstract line 31, increase

line 140  were removed

similarly, lines 124 and 355 include non-sentences- something is missing there.

a general criticism, especially in lines 258-63, is that causation and correlation are not well differentiated. You show correlation. That alone means nothing, except when negative. Please build a better case for your conclusions.

Author Response

Thank you very much for reviewing and commenting on the article. Your suggestions have played a big role in improving the quality of the article. Please see the attachment for specific modification.
